# Different Designs of Deep Marginal Elevation and Its Influence on Fracture Resistance of Teeth with Monolith Zirconia Full-Contour Crowns

**DOI:** 10.3390/medicina59040661

**Published:** 2023-03-27

**Authors:** Ali Robaian, Abdullah Alqahtani, Khalid Alanazi, Abdulrhman Alanazi, Meshal Almalki, Anas Aljarad, Refal Albaijan, Ahmed Maawadh, Aref Sufyan, Mubashir Baig Mirza

**Affiliations:** 1Conservative Dental Science Department, College of Dentistry, Prince Sattam bin Abdulaziz University, Al Kharj 11942, Saudi Arabiam.mirza@psau.edu.sa (M.B.M.); 2Dental Intern, College of Dentistry, Prince Sattam bin Abdulaziz University, Al Kharj 11942, Saudi Arabia; 3Department of Prosthodontics, College of Dentistry, Prince Sattam bin Abdulaziz University, Al Kharj 11942, Saudi Arabia; 4Department of Restorative Dental Sciences, College of Dentistry, King Saud University, Riyadh 11545, Saudi Arabia; 5Department of Dental Biomaterials, College of Dentistry, King Saud University, Riyadh 11545, Saudi Arabia

**Keywords:** computer-aided designing, computer-aided manufacturing, dental crowns, deep marginal elevation, fracture resistance, zirconia

## Abstract

*Background and objectives:* Even with the demand for high esthetics, the strength of the material for esthetic applications continues to be important. In this study, monolith zirconia (MZi) crowns fabricated using CAD/CAM were tested for fracture resistance (FR) in teeth with class II cavity designs with varying proximal depths, restored through a deep marginal elevation technique (DME). *Materials and Methods*: Forty premolars were randomly divided into four groups of ten teeth. In Group A, tooth preparation was conducted and MZi crowns were fabricated. In Group B, mesio-occluso-distal (MOD) cavities were prepared and restored with microhybrid composites before tooth preparation and the fabrication of MZi crowns. In Groups C and D, MOD cavities were prepared, differentiated by the depth of the gingival seat, 2 mm and 4 mm below the cemento-enamel junction (CEJ). Microhybrid composite resin was used for DME on the CEJ and for the restoration of the MOD cavities; beforehand, tooth preparations were conducted and MZi crowns were and cemented using resin cement. The maximum load to fracture (in newtons (N)) and FR (in megapascals (MPa)) were measured using the universal testing machine. *Results:* The average scores indicate a gradual decrease in the load required to fracture the samples from Groups A to D, with mean values of 3415.61 N, 2494.11 N, 2108.25 N and 1891.95 N, respectively. ANOVA revealed highly significant differences between the groups. Multiple group comparisons using the Tukey HSD post hoc test revealed that Group D had greater DME depths and showed significant differences compared with Group B. *Conclusions:* FR in teeth decreased when more tooth structure was involved, even with MZi crowns. However, DME up to 2 mm below the CEJ did not negatively influence the FR. Strengthening the DME-treated teeth with MZi crowns could be a reasonable clinical option, as the force required to fracture the samples far exceeded the maximum recorded biting force for posterior teeth.

## 1. Introduction

Dental caries, or tooth fractures, which extend subgingivally complicate the restorative treatment approach [1]. They are challenging to restore and could result in the violation of the supracrestal connective tissue attachment (STA), leading to gingival inflammation and loss of periodontal supporting tissues [2]. Different techniques and procedures have been used to restore this critical area, keeping in mind the STA [3]. In fact, up to 10% of all periodontal surgical procedures are conducted to increase the crown length [4]. However, osteotomy itself is not without its shortcomings, such as being invasive with reduced patient acceptance, and if is not cautiously carried out, it can alter the crown–root ratio [5].

Deep marginal elevation (DME) is an alternative restorative technique suggested by Dietschi and Spreafico in 1998 [6]. It involves raising the deep margins to supragingival levels, noninvasively, to counter the difficulties encountered with subgingival margins [7]. This is achieved by using special matrix bands of shortened heights, which are secured subgingivally and tightened. Various materials have been presented to elevate the margins; however, DME conducted with composite resin seems to be well tolerated by the periodontium, even allowing the binding of epithelial fibers [8].

When the marginal ridge/s are involved in large lesions, it is wise to plan for indirect restorations (IRs), as the strength of the tooth is directly proportional to the integrity of the marginal ridges [9]. Depending on the extent of tooth structure loss, the need for cuspal coverage and the amount of reinforcement needed, IRs such as inlays, onlays, overlays and full-contour crowns are recommended [10]. Historically, IRs fabricated using gold alloys were considered the gold standard. However, due to increased demand for more natural and esthetic restorations, the use of composites and ceramics have more or less overhauled the use of metals [11].

With the advent of CAD/CAM technology and the ever-evolving choice in esthetic materials, more customizable and predictable restorations are being fabricated, minimizing the laboratory steps or eliminating them completely [12]. Zirconia (Zi) is usually the material of choice when posterior crowns are fabricated through CAD/CAM technology [13,14]. In lieu of the higher chipping rate of veneering porcelains on the Zi copings, full-contour zirconia crowns/monolith zirconia crowns (MZi) were developed [15]. MZi fabricated by CAD/CAM has been proven to be precise, stable and homogenous and has improved qualities compared to its predecessors [15]. Furthermore, Zi has been shown to achieve higher bond strengths if proper connections are achieved with resin adhesives by following the manufacturers’ guidelines regarding surface pretreatment [16,17]. Studies related to the fracture resistance (FR) of teeth with monolith inlays, onlays and endocrowns exist in the literature. However, a dearth of information regarding the effect of DME on the FR of teeth with full-contour crowns led the authors to initiate the present study. Thus, by conducting this research, the authors aimed to evaluate the maximum load and resistance to fracture in teeth with different designs of DME, which were reinforced by MZi crowns. The null hypothesis is that there would be no difference in the FR of DME-treated teeth with MZi crowns.

## 2. Materials and Methods

### 2.1. Study Design

This experimental study was performed in the College of Dentistry, Prince Sattam bin Abdulaziz University (PSAU), Al Kharj, and the College of Dentistry, King Saud Bin Abdulaziz University (KSAU), Riyadh, after obtaining ethical approval from the Institutional Review Board, PSAU, IRB No (PSAU2021011), and the College of Dentistry Research Center, KSAU, CDRC No (FR0626).

### 2.2. Sample Collection

Non-carious maxillary premolars with all cusps and walls intact, freshly extracted for orthodontic reasons, were collected from the specialist clinic, PSAU. These teeth were inspected using a stereomicroscope (RX-100, Hirox, Tokyo, Japan) at 16× magnification for the presence of any deformities, craze lines or indications of fractures, which if found were excluded. Forty teeth which met the inclusion criteria were disinfected with 5.25% sodium hypochlorite for 30 min and stored in normal saline at room temperature until further use.

### 2.3. Description of the Experimental Groups

Selected teeth were divided into four groups, A–D, comprising of ten teeth each, allotted in a random order. All the teeth received full-contour crowns using CAD-CAM technology; however, groups would be differentiated based on the depth of DME from crown margins (Figure 1). Group A: Teeth with full-contour crowns without cavity preparation (control group). Group B: Teeth with mesio-occluso-distal (MOD) preparation/restoration with full-contour crowns (gingival seat 2 mm above the crown margin). Group C: Teeth with MOD preparation/restoration with DME and full-contour crowns (2 mm DME to crown margin). Group D: Teeth with MOD preparations with DME and full-contour crowns (4 mm DME to crown margin). The tooth-preparation scheme is shown in Figure 1.

### 2.4. Tooth Preparation

Orthodontic resin (Ortho-Resin, DeguDent GmbH, Hanau, Germany) was used to cover the roots of the teeth up to 5 mm below the CEJ. The preparation of samples was performed by one investigator to maintain uniformity. The proximal box preparations (mesial or/and distal) were conducted using a using a high-speed contra-angle handpiece Ti-Max Z900L (NSK, Nakanishi Inc., Tochigi, Japan) at a speed range of 320,000–400,000 rpm with a round-end tapered diamond (Bur # TR-14, ISO 198/022, Mani Inc., Tokyo, Japan), which was replaced by a new one for each tooth. The final tooth preparations had the following dimensions: 3 mm wide isthmus; 2 mm wide box at the gingival third and 4 mm occlusal width. The apical depth of the boxes in Group B was prepared 2 mm above the level of CEJ, and for Groups C and D, it was prepared 2 mm and 4 mm below the CEJ, respectively. For etching the dentin, 37% phosphoric acid gel (Prime-Dent, Chicago, IL, USA) was used for 15 s and rinsed with water. The excess water was then carefully removed by a brief burst of air, leaving the dentine and enamel surface slightly moist with a shiny surface. A fully saturated brush was used to apply the bonding agent (Bonding resin, Prime-Dent, Chicago, IL, USA) on the etched surface in two consecutive coats; it was air dried for 10 s and light-cured using an LED curing light (Smart lite max, Dentsply Caulk, Milford, DE, USA) for another 10 s on both the occlusal and proximal surfaces. The rest of the cavity in all samples was filled incrementally using a microhybrid composite, Filtek Z250 (3M ESPE, St. Paul, MN, USA), followed by light curing for 20 s.

Based on current principles of tooth preparation, for all-ceramic zirconia crowns, the occlusal surfaces were reduced by 2.0 mm, with a function-cusp bevel. A circumferential reduction by 1.0 mm was conducted, with the margins terminating at the CEJ. The final preparation had an axial wall taper of 6 to 8 degrees. The preparations were carried out using a flat-end long tapered diamond (Bur # TF-14, ISO 172/023, Mani Inc., Japan) with a highspeed contra-angle handpiece Ti-Max Z900L (NSK, Nakanishi Inc., Tochigi, Japan).

### 2.5. Fabrication of Crowns

All samples were digitally scanned using the Cercon eye scanner (DeguDent GmbH, Hanau, Germany) and designed by the software of Cercon art 3.2 (DeguDent GmbH, Germany) to receive Cercon HT full-contoured crowns. The milling process was accomplished with a Cercon Xpert machine (DeguDent GmbH, Germany) after selecting a zirconia bur and a Cercon disc (DeguDent GmbH, Germany) containing 94.5% pure zirconium. To attain full strength, sintering was accomplished in the dental lab in a Cercon heat plus P8 machine (Dentsply Sirona, NC, USA) set at 8 h and 30 min by a dental technician with a sintering device Cercon Heat (DeguDent GmbH, Germany). The visually unacceptable teeth and those with margin damages were rejected, and another coping was made as needed. The intaglio surface of the Zi crowns were pretreated, mechanically, through air blasting with alumina oxide and cleaned with an alkaline agent, Ivoclean (ivoclar vivadent, Schaan, Liechtenstein), before cementation.

### 2.6. Cementation

The cementation of the crowns was conducted using an A2 shade, dual-cure, self-adhesive resin cement, RelyX U200 (3M, St. Paul, MN, USA), under a constant load of 20 N. The load was applied using a surveyor assembly machine to ensure equal pressure over the crowns, which were then light-cured for 20 secs per surface after cleaning the excess cement beyond the margins.

### 2.7. Aging

The aging of the samples to stimulate clinical conditions was conducted in a thermocycling machine (SD Mechnotronik THE 1100, Feldkirchen-Westerham, Germany). The samples were placed in a 10 × 10 open specimen basket and subjected to 5000 cycles in a water bath at (5 to 55 °C) for 30 s/cycle and a 5 s transfer time.

### 2.8. Fracture Test

Samples were subjected to a fracture test using the MTS 810 Universal Testing Machine (Eden Prairie, MN, USA). After mounting the samples on a metal base at a vertical angle, a stainless-steel flat-load cell was used, making sure that it contacted both the cusps before the force was applied in a vertical direction along the long axis of the tooth (Figure 2). The maximum force until fracture (1 mm/1 min) of the sample was recorded.

### 2.9. Statistical Analysis

ANOVA was used for the statistical analysis of the data. The comparative evaluation of the means was performed using the Tukey HSD post hoc test. A calculated *p*-value of less than 0.05 was considered significant.

## 3. Results

The maximum load-to-fracture values in newtons (N) and fracture-resistance values in megapascals (MPa) for the samples in all groups are listed in Table 1.

To compare and interpret the results of the experimental data seen in Table 1, analysis of variance (ANOVA) was performed using the mean values of each group, as seen in Table 2 and represented as bar diagrams in Figure 3 and Figure 4.

The results showed very high levels of significance among the groups, with the highest load and resistance to fracture seen in the control group, where the Zi crown was cemented on the prepared tooth without any cavity preparation or use of DME. On the other hand, the crowned teeth in Group D, which had MOD restorations and DME with a depth of 4 mm on either side, displayed the least load and resistance to fracture.

In order to compare and assess the significance between individual groups, the Tukey HSD post hoc test was performed, Table 3.

Very high levels of significance were seen when the control, Group A, was compared to all the other groups. Values of Group B also showed a very significant difference compared to Group D. Although the difference in means was noticeable between all the other groups, the results were not of significance.

Table 4 shows the percentages of failed patterns grouped under “restorable” for those samples with the presence of fracture lines or cracks on the Zi crowns, and “unrestorable” for those which failed below the CEJ.

## 4. Discussion

Modern restorative dentistry aims to preserve healthy dentition and replace missing tooth structures conservatively [18]. However, some clinical situations dictate excessive removal, compromising the strength of the remaining tooth structure, especially when the marginal ridges are removed [7,9]. Reeh et al. demonstrated that the tooth loses 63% of its stiffness if MOD preparations are made, and its FR was found to be lower than those teeth in which access opening was performed with intact walls [19]. More recently, a finite-element analysis (FEA) study demonstrated the least resistance to fracture in teeth where both the marginal ridges were removed, re-establishing the concept that the loss of tooth structure adversely affects the strength of the tooth [4]. The consolidation of such teeth should be accomplished by using IRs such as onlays or overlays, and in teeth which are compromised, full-coverage restorations are preferred [20].

Additionally, complete caries removal can also lead to the establishment of deep subgingival margins, which are often difficult to restore [21,22]. Surgical crown lengthening (SCL) is often the recommended treatment in such situations, where the cavity margins are relocated by apically displacing the periodontal attachment [23]. Despite its advantages, SCL is not always acceptable, due to delayed healing, loss of bony support, the elongation of clinical crowns, the development of black triangles and difficulties in estimating the final position of the margins [24].

Studies on the effect of DME on the periodontium indicate that it is well tolerated, both clinically and histologically [25]. In a long-term study comparing the survival rate of teeth, Mugri et al. suggested that the teeth in which DME was performed had a better rate of survival than those in which SCL was performed [24]. Dablanca-Blanco, in a case series, suggested the use of DME in teeth where the subgingival margins are limited in the epithelial tissue attachment, and SCL in teeth where the margins have reached the connective tissue attachment or the crestal bone level [2]. Ghezzi et al. proposed a new classification designed for choosing the line of treatment, based on the ability to isolate the tooth after caries removal and cavity preparation into class 1: nonsurgical DME, class 2a: surgical DME (gingival approach) and class 2b: surgical DME osseous approach [26].

In the present study, a traditional hybrid composite was used for the DME, as it is known to reduce marginal gap formation. The use of other materials such as glass ionomers, resin-modified glass ionomers and smart dentin replacement (SDR) bulk-fill composites for DME have also provided acceptable results with regards to FR, marginal adaptation and microleakage [27,28]. However, contrasting reports on the use of flowable composites have been reported in other studies [29]. The choice in hybrid composite in this study for DME and simultaneous restoration of the cavity was believed to eliminate the development of different interfaces.

The comparison of IRs such as onlays, overlays/partial crowns and full crowns have been extensively studied in the literature, with the majority of these studies reporting them to have clinical efficacy in excess of 93% [20,24]. Full-contour crowns continue to be used widely throughout the world, particularly in teeth which are extensively worn or severely destroyed [30]. A study comparing them to partial crowns found that the full crowns performed better from a technical, biologic and esthetic point of view [31]. Their use is time-tested and successful, regardless of the material and technique used to fabricate them [32]. However, with the use of CAD/CAM technology, monolith crowns fabricated entirely with Zi have gained popularity for use in posterior teeth due to their high strength and esthetics [20]. More recent studies on the clinical service of MZi crowns have shown a clinical success rate of more than 98% [14,33,34].

The FR of posterior teeth was determined previously to be in the range of 1120–2500 N [35]. In this study, the mean value for the maximum load required to fracture the teeth with MZi crowns, without being influenced by the effects of prior cavity preparations or DME, was found to be 3415.613 N (Group A). A 27% reduction in load to fracture was seen in Group B, which had mean values of 2494.118 N. This could be explained by the reduction in strength seen in teeth due to the effects of MOD cavity preparation/restoration. However, unlike in the teeth without crowns, where the effects of MOD preparations are severe, a less drastic reduction in strength was noticed due to the use of MZi crowns, which are known to provide cushioning and protection to the underlying tooth structures [36]. The mean values in this study were lower than those described by Nakamura et al., who reported that a load of 5000 N was required to fracture the crowns, slightly higher when than in another study which reported a mean value of 2337 N [37,38]. These differences could be due to the use of molar dyes fabricated using a hybrid polymer resin in the study by Nakamura, who also reported that the heights of axial walls could influence the FR of teeth under all ceramic crowns [37].

Research on the effects of DME on FR among indirect CAD/CAM restorations have generally focused on teeth with inlays, onlays, overlays and endocrowns. Most of these studies reported that the use of DME did not significantly impact the FR of teeth [21,22,39]. Grubbs et al., comparing the use of 2 mm-deep DME to CEJ in MOD cavities restored with onlays, displayed mean FR values between 1700 and 2029 N, which are slightly lower than in our study [28]. Furthermore, Zhang et al. demonstrated improved FR on root-canal-treated teeth, restored with CAD/CAM endocrowns following the use of DME [27,40]. Despite many such studies on IRs, the effect of DME on teeth under full-contour crowns have rarely been reported. Only one study could be partially compared to our study, in which the authors used CAD-/CAM-based monolith lithium–disilicate crowns over MOD prepared/restored premolars with DME [35]. The results showed fracture forces in the range of 1638–2253 N, comparable to values achieved in the experimental groups of this study where cavities (Groups B-D) were prepared, 1891.956–2494.118 N [41].

The type and site of fracture was visually evaluated. Failure type was classified based on the adaptation and modification from previous studies, as “restorable” for those with visible cracks and fracture lines on the zirconia crowns and “unrestorable” for those in which the fractures were seen below the CEJ [27,42,43]. In most of the samples in the present study, the teeth were unrestorably fractured below the CEJ. The results were expected due to the fact that MZi crowns usually fracture at higher loads and present with complete cohesive fractures, unlike the veneered zirconia crowns, which tend to chip [15]. Similar high percentages of unrestorable failure were seen in other in vitro studies [28,41,44]. The rationale for this could be the transmission of forces to the underlying natural teeth abutments, resulting in their fractures at high loads [15]. However, the choice in natural teeth in the present study was necessary to evaluate the effects of DME on FR, to better replicate the clinical scenario based on adhesive characteristics, strength and modulus of elasticity. The occurrence of restorable failures seen in the study could be related to fabrication defects compromising the integrity of MZi [45].

Under the limitations of this study, DME under normal depths (2 mm) rarely influenced the FR of teeth, except when the tooth structure was severely compromised, as in Group D. However, the load required to fracture the crowned teeth decreased as the extension of DME apical to the CEJ increased, thus rejecting the null hypothesis. Most of the samples failed unfavorably under maximum load. Nevertheless, more importantly, from a clinical point of view, the load required to fracture these samples far exceeded the maximum recorded biting force in posterior teeth, which is in the range of 800–1000 N [15].

In spite of the perceived advantages in this study, it has some limitations; a. The ease in performing the DME, especially where the distance to the crown margins is greater than 2 mm, might not be established in clinical settings without STA violations. b. Natural teeth variances in strength and morphology when compared to the dyes used in some of the other studies could have an influence on the results. c. The fracture of the teeth/MZi crown assembly was evaluated visually in this study; a more detailed fractographic evaluation would have provided precise details regarding the initiation and causes of cracks. The future trends in research will change the ever-evolving material choices for direct restorations and the development of monolith crowns using newer CAD/CAM technologies, which will help establish an ideal restorative choice, esthetically and functionally.

## 5. Conclusions

The loss of tooth structure, especially the marginal ridges, adversely affects the strength of teeth, even if supported by high-strength full-contour Zi crowns. The DME technique is suitable for use when deep margins are encountered, and the length of the DME-to-crown margins does not significantly impact its fracture resistance. However, caution should be undertaken in MOD preparations where the distance between the DME and the crown margins reaches 4 mm. Although unfavorable failures were common in this study, the load required to fracture all the samples, regardless of the DME margins, far exceeded the maximum recorded biting force.

## Figures and Tables

**Figure 1 medicina-59-00661-f001:**
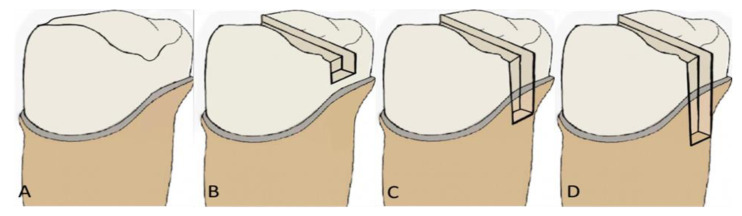
Schematic representation of groups. Group (**A**): Teeth with full-contour crowns without cavity preparation (control group). Group (**B**): Teeth with mesio-occluso-distal (MOD) preparation/restoration with full-contour crowns (gingival seat 2 mm above the crown margin). Group (**C**): Teeth with MOD preparation/restoration with DME and full-contour crowns (2 mm DME to crown margin). Group (**D**): Teeth with MOD preparations with DME and full-contour crowns (4 mm DME to crown margin).

**Figure 2 medicina-59-00661-f002:**
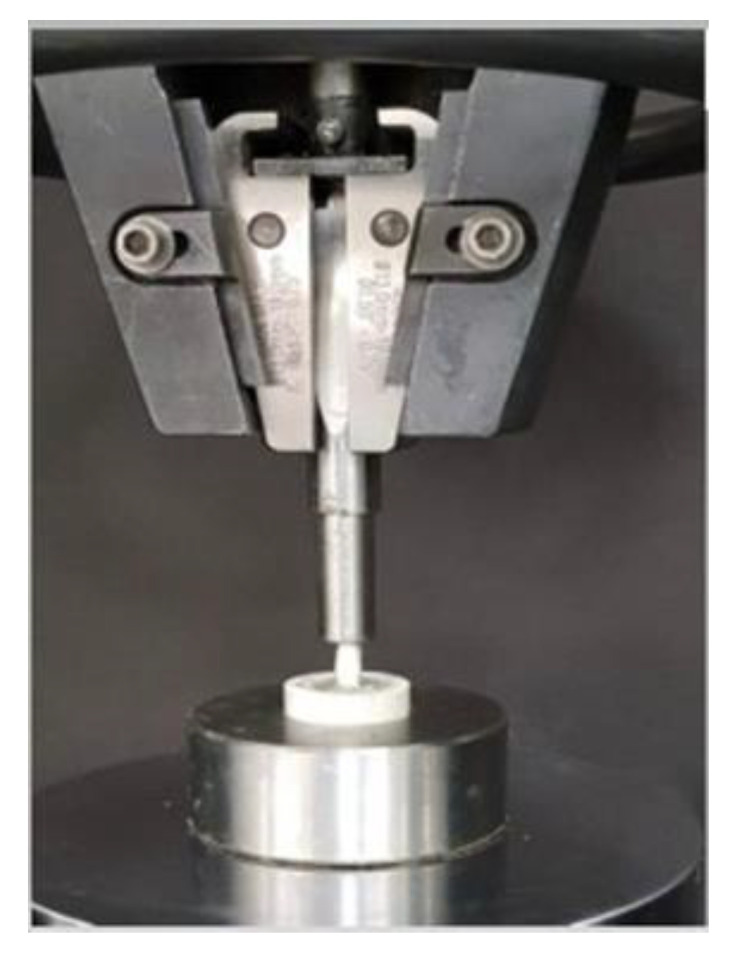
Fracture resistance test using Universal Testing Machine.

**Figure 3 medicina-59-00661-f003:**
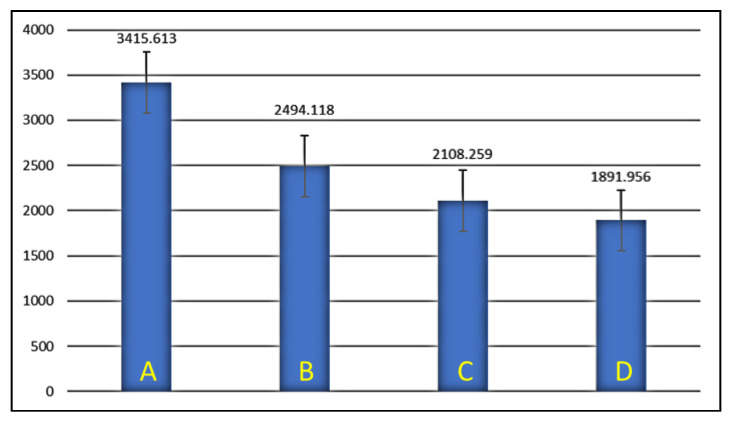
Mean values of maximum load to fracture expressed in newtons (N). A, Group A; B, Group B; C, Group C; D, Group D.

**Figure 4 medicina-59-00661-f004:**
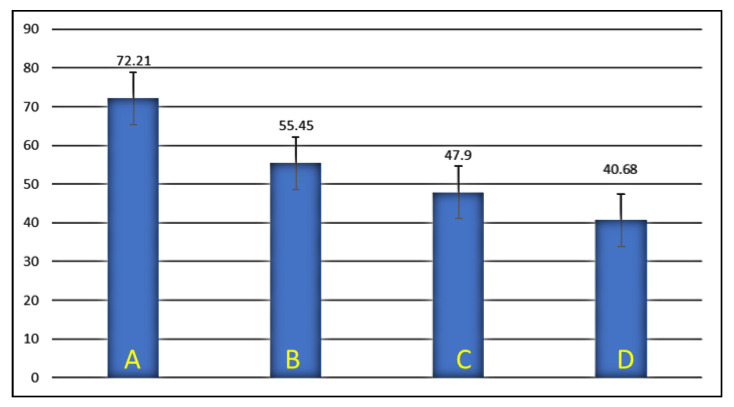
Mean values of fracture resistance in megapascals (MPa). A, Group A; B, Group B; C, Group C; D, Group D.

**Table 1 medicina-59-00661-t001:** Maximum load and fracture-resistance scores of samples in different groups.

Sample	Control		Mesio-Occluso-Distal (MOD) Tooth Preparation					
	A		B		C		D	
	Maximum Load (N)	FR (MPa)	Maximum Load (N)	FR (MPa)	Maximum Load (N)	FR (MPa)	Maximum Load (N)	FR (MPa)
1	3436.55	72.65	2040.66	56.06	1460.13	34.97	1368.39	30.23
2	3586.65	75.82	2072.61	45.22	2012.64	42.62	2285.31	40.88
3	3479.02	73.55	1801.64	49.57	2253.41	51.77	2168.52	45.84
4	3219.50	68.06	2222.84	48.38	2704.48	57.17	2103.77	41.80
5	3357.72	70.98	2959.94	62.57	2173.11	45.94	2180.15	43.61
6	3485.54	73.69	3170.40	67.02	2452.58	51.85	1358.02	32.16
7	3411.18	72.11	2737.58	57.87	1582.64	47.36	1883.83	48.56
8	3393.20	71.73	2654.50	56.12	2014.36	42.58	1979.91	45.99
9	3436.22	72.64	2773.51	58.63	2597.09	54.90	1854.38	41.00
10	3350.55	70.83	2507.50	53.01	1832.15	49.87	1737.28	36.72

N, Newtons; FR, Fracture resistance; MPa, Megapascals. Group A: Tooth preparation to receive full-contour crowns. Group B: Tooth preparation to receive full-contour crowns and mesio-occluso-distal (MOD) cavity preparation. Group C: Tooth preparation to receive full-contour crowns and MOD cavity preparation with DME (2 mm DME to crown margin). Group D: Tooth preparation to receive full-contour crowns and MOD tooth preparation with DME (4 mm DME to crown margin).

**Table 2 medicina-59-00661-t002:** Comparison between groups using one-way ANOVA.

Units	Groups	*n*	Mean	Std. Deviation	95% Confidence Interval		*p*-Value
					Lower Bound	Upper Bound	
Max load(N)	A	10	3415.61	97.52	3345.8503	3485.3757	<0.00001 ***
	B	10	2494.11	443.97	2176.5202	2811.7158	
	C	10	2108.25	412.15	1813.4246	2403.0934	
	D	10	1891.95	325.14	1659.3600	2124.5520	
FR (MPa)	A	10	72.21	2.06	70.737	73.687	<0.00001 ***
	B	10	55.45	6.63	50.701	60.199	
	C	10	47.90	6.64	43.157	52.659	
	D	10	40.68	6.00	36.391	44.979	

***, Very high levels of significance with *p* value ≤ 0.05.

**Table 3 medicina-59-00661-t003:** Dependance of variable; Tukey HSD intergroup comparison.

Groups	Comparative Groups	Mean Difference	*p*	95% Confidence Interval	
				Lower Bound	Upper Bound
A	Group B	921.49500	<0.001 ***	465.0409	1377.9491
	Group C	1307.35400	<0.001 ***	850.8999	1763.8081
	Group D	1523.65700	<0.001 ***	1067.2029	1980.1111
B	Group C	385.85900	0.133	−70.5951	842.3131
	Group D	602.16200	0.004 **	145.7079	1058.6161
C	Group D	216.30300	0.664	−240.1511	672.7571

*** very highly significant; ** highly significant; with *p* value ≤ 0.05.

**Table 4 medicina-59-00661-t004:** Failure patterns.

Groups	*n*	R (%)	U (%)
A	10	1 (10)	9 (90)
B	10	1 (10)	9 (90)
C	10	1 (10)	9 (90)
D	10	2 (20)	8 (80)

*n*, number of samples; R, restorable, with crack lines in the Zi crowns; U, unrestorable, with fractures below the CEJ.

## Data Availability

Available on request.

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
