# Peer review of "Different Designs of Deep Marginal Elevation and Its Influence on Fracture Resistance of Teeth with Monolith Zirconia Full-Contour Crowns"

_medicina, 2023, doi:10.3390/medicina59040661_

Round 1

Reviewer 1 Report

Very interesting article. Discussing important issues such as not only aesthetic reconstruction with crowns made of zirconium oxide, but factors affecting the long clinical period of this type of restorations. However like a reviewer I have some comments how to improve this work:

Abstract.

MOD, CEJ- Please explain the abbreviations used here. Many readers read the abstract, if they find it interesting, they read the entire article and may quote it. And we're all about citing our work.

It would be worth giving numerical values between groups A-D in terms of resistance to fracture. This is always good for the reader's reception.

  There are no applications in the abstract. Why this work was done and what it brings to the clinical practice of making zirconia crowns. After all, research is to serve the clinician so that they can work better with materials, and not make in vivo errors that have already been detected in laboratory tests.

Introduction

 Line 53- please explain the aberration IR?

In the production part itself, it would be good to add some information about the proper connections between materials, because it also affects the durability for example:

Raszewski, Z.; Brzakalski,D.; Derpenski, Ł.; Jałbrzykowski, M.;Przekop, R.E. Aspects and Principles of Material Connections inRestorative Dentistry—A Comprehensive Review. Materials 2022, 15, 7131. https://doi.org/10.3390/ma15207131

Thammajaruk P, Inokoshi M, Chong S, Guazzato M. Bonding of composite cements to zirconia: A systematic review and meta-analysis of in vitro studies. J Mech Behav Biomed Mater. 2018 Apr;80:258-268. doi: 10.1016/j.jmbbm.2018.02.008. Epub 2018 Feb 8. PMID: 29454279.

 Materials and methods

Line 84- a stereomicroscope- name of the instrument, producer, country

regular saline - have the teeth been disinfected in some way (possibility of cross-contamination) Storage in this medium without preservatives can cause the growth of microorganisms which in turn can affect the structure of the enamel and the bonding of the material. This would require clarification, Thank you.

The article contains Figure 1, but there is no reference to this picture in the text. In my opinion, I would add a sentence on line 97: The tooth preparation scheme is shown in Figure 1.

The Figure is nice, good job!

What instruments were used to grind these teeth, the company's micromotor, speed of the instrument, the country of origin, etc.

 Line 150 at (5 to 550 C) for 30 seconds/cycle- 55oC.

Results

Table 1. Please enter the units for columns A-D.

Table 2. The results of crush resistance are given with an accuracy of 0.01 N so AVG is Sd also with the same accuracy (instead of 0.001N) yes for mathematical order.

Discussion

 Line 202- SCL , please explain this aberration

Line 210 Murgi or Mugri as you have at 20 of Ref?

Iine 220- SDR???

Line 228 - .16,20 please add [16,20].

Very nice discussion, good done !

You are using a lot of abbreviations at work, add a table at the end of the article where you explain all the abbreviations again. It will help with reading, please.

Good luck with your further research

Reviewer 2 Report

Introduction
-please explain at the first abbreviation use (IR for example)
-no null hypotheses presented, aims are not clearly stated, difficult to be understandable
-no literature support on the increasing use of zirconia claim here

M&M
The study design presents major issues which lowering the significance of the outcomes
-no thermocycling has been implied before crown positioning over tooth. This means that all cavities have freshly adhered restorations which mechanically deform along dentin and mutually reinforcing core.
-Zirconia is retended on mechanical basis, hence is deformed and accepting load separately from teeth. This is a completely different behaviour versus any material positioned with adhesion on teeth tissues. No information has been given on where and how fractures happened. There is a vast difference between a fracture of the cervical part of the tooth versus a fracture of the material. Maximum load force alone does not provide any useful information.
-Forces expressed in N are not comparable. There should be MPa utilized here. Of course this is related to sample area and since natural teeth have been used there should be a big deviation on results on maximum load force. However comparing Newton force is useless, an area calculation could be possible after scanning specimens.
-No information on loading direction provided
-No details on storage solution (natural saliva? provided by whom? in what temperature? how was bacterial development controlled?)

Results
a basic analyse on maximum load forces, can not lead to any conclusion

Conclusions
I would expect at least a conclusion on type of fracture and if this might be a factor against tooth integrity or restoration
